# In Vitro Antifungal Activity of Chimeric Peptides Derived from Bovine Lactoferricin and Buforin II against *Cryptococcus neoformans* var. *grubii*

**DOI:** 10.3390/antibiotics11121819

**Published:** 2022-12-15

**Authors:** Silvia Katherine Carvajal, Yerly Vargas-Casanova, Héctor Manuel Pineda-Castañeda, Javier Eduardo García-Castañeda, Zuly Jenny Rivera-Monroy, Claudia Marcela Parra-Giraldo

**Affiliations:** 1Unidad de Proteómica y Micosis Humanas, Grupo de Enfermedades Infecciosas, Departamento de Microbiología, Facultad de Ciencias, Pontificia Universidad Javeriana, Bogotá D.C. 110231, Colombia; 2Chemistry Department, Universidad Nacional de Colombia, Carrera 45 No. 26–85, Building 451, Office 409, Bogotá D.C. 111321, Colombia; 3Pharmacy Department, Universidad Nacional de Colombia, Bogotá Carrera 45 No. 26–85, Building 450, Bogotá D.C. 111321, Colombia

**Keywords:** *Cryptococcus neoformans*, cryptococcosis, antimicrobial peptides, chimeric peptides, antifungals, buforin II, bovine lactoferricin

## Abstract

Cryptococcosis is associated with high rates of morbidity and mortality. The limited number of antifungal agents, their toxicity, and the difficulty of these molecules in crossing the blood–brain barrier have made the exploration of new therapeutic candidates against *Cryptococcus neoformans* a priority task. To optimize the antimicrobial functionality and improve the physicochemical properties of AMPs, chemical strategies include combinations of peptide fragments into one. This study aimed to evaluate the binding of the minimum activity motif of bovine lactoferricin (LfcinB) and buforin II (BFII) against *C. neoformans* var. *grubii.* The antifungal activity against these chimeras was evaluated against (i) the reference strain H99, (ii) three Colombian clinical strains, and (iii) eleven mutant strains, with the aim of evaluating the possible antifungal target. We found high activity against these strains, with a MIC between 6.25 and 12.5 µg/mL. Studies were carried out to evaluate the effect of the combination of fluconazole treatments, finding a synergistic effect. Finally, when fibroblast cells were treated with 12.5 µg/mL of the chimeras, a viability of more than 65% was found. The results obtained in this study identify these chimeras as potential antifungal molecules for future therapeutic applications against cryptococcosis.

## 1. Introduction

Invasive fungal infections are on the rise, causing high morbidity and mortality rates in immunocompromised individuals, especially those with lymphocytes below 200 cells/mm^3^, including HIV patients, cancer patients receiving chemotherapy, individuals who have undergone solid organs transplants, and those with hematopoietic stem cell transplantations [1,2,3,4].

*Cryptococcus neoformans* is an opportunistic basidiomycete encapsulated fungus that can cause cryptococcosis, especially in pulmonary regions or even in the meninges [5]. There are two varieties of *C. neoformans*: *grubii* (serotype A), which is ubiquitous worldwide, and *neoformans* (serotype D), found mainly in Europe [6]. The most prevalent species is *C. neoformans*, *grubii* being the most frequent variety. Therefore, the present study focuses on *C. neorformans* var. *grubii* [7].

Cryptococcosis is a potentially fatal opportunistic mycosis of global incidence. The global incidence of cryptococcal meningitis is estimated to be 223,100 cases per year, with 73% of the cases in sub-Saharan Africa. Worldwide, cryptococcal meningitis accounts for 181,100 deaths per year and is responsible for approximately 15% of AIDS-related deaths [8]. The incidence of cryptococcosis in Colombia in the period 1997–2016 was one case per 434,782 people. The departments with the highest incidence were Norte de Santander (1 case per 178,571 people) and Valle (1 case per 212,766 people) [9].

Despite established recommendations for the treatment of cryptococcosis, there are several limitations related to the antifungals that currently are clinically available: their global access remains restricted; there are few therapeutic alternatives; security profiles impose limitations; and drug penetration and distribution to the site of infection is challenging [10,11,12,13]. In addition, it has been described that *C. neoformans* can develop heteroresistance to fluconazole [14].

As a result of the problems described above, a demand for the development of new antifungal molecules has been triggered. Antimicrobial peptides (AMPs) have emerged as pharmaceutical candidates with promising antifungal activity. AMPs are molecules that are widely distributed in nature and play an important role in the innate immune system of various organisms. These peptides exhibit broad-spectrum antimicrobial and immunomodulatory activities against various microorganisms, including yeasts and molds. Most AMPs are cationic and amphipathic. Cationic AMPs are often helical, with short amino acid sequences (less than 100 amino acid residues) [15].

Antimicrobial peptides derived from bovine lactoferricin (LfcinB) and buforin II (BFII) have been shown to exhibit strong antimicrobial activity not only against bacteria but also against yeasts such as *C. neoformans* [16,17]. Synthetic chimeric peptides have emerged as a strategy for improving and enhancing the antimicrobial effects of AMPs. In a 2021 study, these chimerical peptides demonstrated a high degree of activity against both Gram-negative and Gram-positive bacteria [18]. Another recently published study tested these chimeras on *Candida* spp. [19]. The antifungal effect of these chimeras against *C. neoformans* has not been reported.

In light of the above, the present study evaluated the in vitro antifungal activity of chimeric peptides containing LfcinB (20–25): RRWQWR and BFII (32–35): RLLR sequences; specifically, the effect of chimeras on *C. neoformans* var. *grubii* strains.

## 2. Results

### 2.1. Antifungal Activity of Chimeric Peptides Derived from Bovine Lactoferricin and Buforin II against Cryptococcus neoformans var. grubii

The designed and synthesized peptides used for the purpose of this study were: C5: RRWQWR-Ahx-KLLKKLLK, C6: KKWQWK-Ahx-RLLRRLLR, and precursor peptides LfcinB (20–25): RRWQWR, BFII (32–35)_Pal_: RLLRRLLR, [K]-LfcinB (20–25): KKWQWK, and [K]-BFII (32–35)_Pal_: KLLKKLLK [18,19]. In both chimeras, 6-aminohexanoic acid (Ahx) was used as the spacer molecule. The H99 strain and one clinical isolate were sensitive to fluconazole (FLC), and two clinical isolates were considered dose-dependent sensitive (DDS) to this antifungal (Table 1). All mutants were sensitive to FLC (Table 2). For peptide C5, activity was found with a MIC of 12.5 µg/mL and an MFC of 12.5 µg/mL, depending on the strain; peptide C6 exhibited activity against the strains evaluated with a MIC of 6.25 µg/mL and an MFC of 6.25 µg/mL. No antifungal activity of the precursor peptides was observed at the concentrations tested against *C. neoformans* var. *grubii*.

The antifungal activity of chimeras and their precursor sequences in mutants involved in the TOR signaling pathway, ergosterol biogenesis, efflux pump mechanisms, and cell cycle control was evaluated. Mutants exposed to the C5 chimera exhibited a MIC between ≤3.125 and 12.5 µg/mL; when the mutants were incubated with the C6 chimera, they exhibited a MIC of 3.125 to 6.25 µg/mL. On the other hand, when treating the mutants with the precursor peptides of LfcinB (20–25) and BFII (32–35)_Pal_, the MIC was >25 and >50 µg/mL, respectively (Table 2).

### 2.2. Growth Inhibition and Killing Kinetics

The growth inhibition kinetics of the chimeras were evaluated against *C. neoformans* at different time intervals. The inhibitory effect of chimeras began 14 h after the start of incubation. With increasing time, the inhibitory effect of low concentrations of both chimeras on the fungal growth curve weakened. However, they still exhibited a significant inhibitory effect when compared with the precursor peptides. When the concentration of peptide C5 was 12.5 µg/mL, an ~30% decrease in growth was observed at 68–72 h (Figure 1a), while the fungicidal effect was observed at 25 µg/mL. The H99 strain was incubated with chimera C6 at 3.125 µg/mL and exhibited a prolongation of the lag phase (adaptation) with a growth reduction of ~27%. With ≥6.25 µg/mL, C6 exhibited a fungicidal effect at 68–72 h (Figure 1b). The 2807 clinical isolate incubated with peptide C5 at a concentration of 3.125 µg/mL exhibited a growth reduction of ~34%. When the peptide concentration was 6.25 to 12.5 µg/mL, a fungistatic effect was observed (reduced growth by 66% and 73%, respectively) (Figure 1c). When the 2807 was incubated with chimera C6 at a concentration of 1.56 µg/mL, it exhibited a growth reduction of ~24%, and when the peptide concentration was 3.125 to 6.25 µg/mL, a fungistatic effect was observed at 68–72 h (reduced growth of 70% and 74%, respectively) (Figure 1d). The 3279 strain incubated with C5 at concentrations of 12.5 µg/mL exhibited a growth reduction of almost 18%, and the fungicidal effect was observed at 25 µg/mL (Figure 1e). When it was incubated with chimera C6 at a concentration of 6.25 µg/mL, it exhibited a growth reduction between ~18% and at a concentration of 12.5 µg/mL, a fungistatic effect was observed at 68–72 h (growth reduction ~76%) Table 3 (Figure 1f).

The 2643 clinical isolate incubated with C5 at concentrations of 3.125 and 6.25 µg/mL exhibited a growth reduction of ~35%; when the concentration was increased to 12.5 µg/mL, a growth reduction of 61% was observed (fungistatic effect) (Figure 1g). When this clinical isolate was incubated with chimera C6 at 1.56 µg/mL, it exhibited a growth reduction of ~20%; at concentrations of 3.125 and 6.25 µg/mL, a fungistatic effect was observed at 68–72 h (Figure 1h).

### 2.3. Fluorescent Staining for Yeast Viability

The antifungal effect of chimera C6 was evaluated at concentrations between 3.125 and 25 μg/mL. A live control without treatment and a control of dead yeast cells treated with hypochlorite were used (results not shown). Live control cells were labeled with a blue fluorescence (calcofluor white M2R), highlighting cell wall chitin regardless of the metabolic state. Yeasts were also labeled with FUN1, using two different excitation filters (488 and 532 nm), revealing a yeast with a diffusely distributed green and red color, indicating plasma membrane integrity and the yeast cells’ metabolic capability to convert this intracellular staining to a red-orange fluorescence (Figure 2).

Although metabolically active structures were observed, after 72 h of incubation with the chimera at concentrations of 6.25, 12.5, and 25 μg/mL, there was a >99% decrease in cell population. When the cells were treated with the peptide at a concentration of 3.125 μg/mL, a cellular decrease was visually observed in relation to the metabolism pattern of the living control. Similarly, a morphological variation was shown compared with the live control. When the cells were treated with fluconazole at 4 μg/mL (corresponding to the MIC), a decrease in the cell population was observed with respect to the live control. When the cells were treated with 8 μg/mL of fluconazole, a cell decrease of >80% was observed.

### 2.4. Checkerboard Assay

Considering that both peptide chimeras showed promising in vitro antifungal activity because the strains evaluated exhibited hypersensitivity to them, we proceeded to study the effect of the combination of each chimera with FLC via the checkerboard method, using the H99 reference strain and clinical isolate 2807. The results showed that the combination of C5 and FLC exhibited an additive effect (∑FIC 1.0) in H99, while the combination of chimera and FLC in clinical isolate 2807 exhibited a synergistic effect, with an ∑FIC index between 0.3 and 0.5 at two concentrations below the MIC of FLC and up to four concentrations below the MIC of this chimera. The combined effect of peptide C6 and FLC in reference strain H99 exhibited additivity, with decreased growth ≥90%, in two concentrations below the MIC of this chimera and one of FLC. For clinical isolate 2807, the analysis showed synergy, with an ∑FIC index between 0.3 and 0.5 up to four concentrations below the MIC of C5 and up to two concentrations below the MIC of FLC (Table 4).

The precursor peptides LfcinB (20–25): RRWQWR, BFII (32–35)_Pal_: RLLRRLLR, [K]-LfcinB (20–25): KKWQWK, and [K]-BFII (32–35)_Pal_: KLLKKLLK did not show a synergistic effect when they were physically mixed, and no significant inhibitory activity was observed at the concentrations tested. When peptides [K]-LfcinB (20–25) (50 to 200 µg/mL) and BFII (32–35)_Pal_ (50 to 100 µg/mL) were combined, an inhibitory effect of approximately 90% was observed (Appendix A).

### 2.5. Cytotoxicity: MTT Assay

The viability of L929 murine fibroblast cells was determined after treatments with chimeric peptides at concentrations between 3.125 and 200 µg/mL. The results indicated that the fibroblasts’ viability decreases depending on the concentration of the chimera, with an IC50 value of 93.35 µg/mL for chimera C5 and 67.44 µg/mL for chimera C6. The viability was almost 10% when the cells were treated with the maximum concentration of chimera C6 (Figure 3).

## 3. Discussion

This is the first study testing chimeric peptides derived from LfcinB and BFII in *C. neoformans*. Our results showed that both chimeras exhibited fungicidal and fungistatic antifungal activity against these yeasts. Pineda, H., et al. evaluated the potential of these molecules (sequences with partial substitution of Arg with Lys) in Gram-negative and Gram-positive bacteria, where the average MICs were 12ؘ–97 µM and 24–48 µM, respectively [18]. Furthermore, they evaluated a chimera derived from LfcinB and BFII with total substitution of Arg with Lys and found a decrease in antimicrobial activity in the bacteria evaluated (MIC: 25–101 µM) [18]. They showed that chimeras with partial substitution of Arg with Lys maintained antimicrobial activity. In addition, it has been proven that the partial substitution of amino acids favors peptide synthesis and reduces costs [20,21].

For the design and synthesis of the peptide chimeras, the minimal antimicrobial motifs reported for each Lfcin B (RRWQWR) or BFII (RLLR) were used. These motifs are the shortest sequence of the original PAM that presents activity in various microorganisms. Park, C.B., et al. synthesized a series of buforin II analogues and evaluated their antifungal activity, showing that the RLLR motif had potent antimicrobial activity [17]. Furthermore, studies have shown that the lactoferricin B active site and minimal motif includes the RRWQWR sequence [22,23].

Our chimeras were designed as follows: (a) contains a partial substitution of Arg with Lys; (b) the LfcinB motif is in the N-terminal position and BFII in the C-terminal position; (c) the chimeras are separated by the spacer Ahx; and (d) a palindromic sequence derived from BFII (RLLRRLLR) was included. Our results indicate that these chimeras exhibit greater activity against all *C. neoformans* strains (≤3.125 to 12.5 µg/mL) compared with the precursor sequences (>25 µg/mL).

Positively-charged amino acid residues, such as Arg and Lys, have been described as playing an important role in antimicrobial activity because they promote electrostatic interactions between peptides and the membranes of microorganisms [24]. Despite this, the chemical differences in the side chain groups for Arg (guanidine group) and Lys (amine group) confer different properties on the peptides, affecting their antimicrobial activity. In most cases, the presence of Lys has been associated with a reduction in the antimicrobial activity of peptides. This behavior can be explained by the higher number of hydrogen bonds that the guanidine group can form in contrast to the amine group [25]. 

The chimeras contain a sequence that has been associated with membrane destabilization and cell lysis (LfcinB (20–25)) and another that can translocate within the cell without membrane damage (BFII (32–35)_Pal_) [26,27]. According to reports, for the design of chimeras, it could be considered that one of the sequences has the capacity for cellular internalization; this approach proposes a strategy to improve antimicrobial activity [28]. The results of this research showed that the chimeras increased the antifungal activity against *C. neoformans* var. *grubii* compared with the precursor peptides.

6-Aminohexanoic acid (Ahx) is a non-peptidic molecule. It exhibits hydrophobic behavior and has a flexible structure and adequate space between its terminal amino and carboxyl groups that give this molecule the possibility of being used as a spacer and linker of fragments of active molecules. Ahx does not necessarily generate antimicrobial activity by itself, but when conjugated with peptides, it can significantly improve antimicrobial activity [29,30].

Our findings show antifungal activity of chimeras in mutants with defects in proteins involved in the TOR signaling pathway, ergosterol biogenesis, efflux pump mechanisms, and cell cycle control (see Appendix A). Using strains with the specific mutations described above, it was proposed that possible cellular mechanisms (modes of action or resistance) that were affected by the action of chimeric peptides could be detected, but this was not possible because resistance to chimeras was not found in mutants. However, other mutants—for example, those that are related to pathways involved in capsule biosynthesis—should be explored [31]. The antifungal activity of the chimeras against the mutants was similar to that of those observed in the reference and clinical strains. All mutants were sensitive to the ergosterol biosynthesis inhibitor used for this study, fluconazole, especially one mutant, *sre1Δ* (CNAG_04804). Sre1 plays a central role in adaptation to low-oxygen growth. Under low-oxygen conditions, oxygen-dependent sterol synthesis is inhibited, leading to Sre1 activation. In *C. neoformans*, Sre1 and Scp1 mediate the adaptation necessary for the growth of this yeast in the host and the progression of cryptococcal infection [32,33]. In our study, we did not test hypoxic or anaerobic conditions. The correlation between chimera antifungal activity and ergosterol synthesis should be evaluated, possibly using mutants directly related to this process, such as *ERG11* (the target of azole antifungal agent mutants) [34,35].

According to the criteria of fungicidal activity (molecules that decrease fungal growth ≥99%) and fungistatic activity (molecules that decrease fungal growth <99%) in the results of the growth kinetics, we determined that these chimeras exhibit fungistatic activity and concentration-dependent fungicide in all of the strains and isolates evaluated. This approach was carried out using two different methodologies, broth microdilution and automated growth kinetics. Upon comparing the results of these two methodologies, few differences were seen. Despite this, it was observed that at concentrations of >12.5 µg/mL, no growth was evidenced in most of the strains when they were treated with the chimeras.

Since the antifungal activity of the two chimeras was similar, it was decided to evaluate cell viability in the reference strain using one of the two chimeras by means of confocal microscopy. For this reason, the C6 chimera was evaluated in the H99 strain by staining with FUN1 and calcofluor white. Interestingly, the analyzed peptide exhibited antifungal activity with total inhibition, in visual terms, at 72 h, when treated with 6.25, 12.5, and 25 µg/mL. On the basis of these results, we were able to corroborate the fungicidal effect exerted by this chimera on yeast cells at concentrations greater than 6.25 µg/mL for the H99 strain.

Combination therapy can be a therapeutic alternative for increasing the antimicrobial effect of drugs. The synergistic effect of the chimeras with FLC was evaluated against strain H99 and a clinical isolate DDS to FLC. The results showed that for H99, the combination chimera/FLC exhibited an effect of additivity and indifference (∑FIC = 1) and indifference (∑FIC = 0.8).

In clinical isolation, both chimeras with FLC showed a strong synergistic effect (∑FIC 0.4 and 0.3, respectively). These findings emphasize the importance of reducing the concentrations of the treatments in order to enhance the activity against *C. neoformans* var. *grubii.* In the H99 strain, both chimera antifungal activities increased by a factor of 2 (MIC_a_/A) when it was combined with 2 μg/mL of FLC. In the 2807 isolate, the antifungal activity of both chimeras increased by a factor of 4 (MIC_a_/A) when they were combined with 4 μg/mL of FLC. In this case, its being a clinical isolate, this increase in activity is interesting because the amount of azole antifungal and the peptide required to kill the yeast decreased.

Due to the high antifungal activity observed for the chimeras, it was decided to evaluate whether the minimal motifs they contained (precursors), that is, the peptides with and without substitution of Arg with Lys, could have a synergistic effect through physical mixing. No activity was observed at the concentrations tested (3.125–100 µg/mL) nor were synergistic effects observed, particularly when mixing the RRWQWR and KLLKKLLK sequences. Regarding the combination of KKWQWK and RLLRRLLR, >90% inhibition of yeast cells was observed when combining peptide KKWQWK at a concentration of 50, 100, and 200 µg/mL with RLLRRLLR at 50 and 100 µg/mL. Although these combinations exhibited an inhibitory effect, it was not comparable to the activity achieved by chimeras. In this context, these results confirm that chimera arrays are necessary for antifungal activity, since they exhibit a strong effect against *C. neoformans* var. *grubii* [18,36].

One of the main concerns about the use of antimicrobial peptides in systemic treatments is their toxicity. With antifungal and broad-spectrum capabilities, the new molecules must exhibit low toxicity to cells before considering medical use. These chimeras exhibited a low hemolytic effect, indicating that the chemical binding of the precursors confers selectivity for bacterial and *C. neoformans* var. *grubii* strains [18]. In a study in 2005, the effect of LfcinB at 200 μg/mL on normal human T lymphocytes, fibroblasts, and endothelial cells was analyzed, demonstrating that this molecule was not toxic for these cultures [37]. Takeshima, K., et al. measured the cytotoxic activity of a BFII derivative against human fibroblast cells and found that it was practically non-toxic, at least up to 100 μM [38]. In our study, an IC50 of 93.35 and 67.44 µg/mL was detected for peptides C5 and C6, respectively, and a viability of >65% was observed when cells were treated with 12.5 µg/mL of each chimera.

## 4. Materials and Methods

### 4.1. Fungal Isolates and Culture Conditions

A total of 15 *C. neoformans* strains were used in this study: (i) H99 reference strain, (ii) 3 Colombian clinical isolates, (iv) 11 mutants (associated with a target protein of subunit 2 of rapamycin complex, enzymes involved in ergosterol biogenesis, proteins with action in efflux pump mechanisms, and cell cycle control) (see Appendix A). The majority of them were chosen on the basis of resistance-related phenotypes and antifungal targets using the FungiDB web interface and the strains’ availability in the collection (Madhani, 2007). 

### 4.2. Compounds

The research group Síntesis y Aplicación de Moléculas Peptídicas (SAMP) synthesized the chimeric and precursor peptides using the solid phase peptide synthesis methodology (SPPS) (Table 1) [18]. The characterization of the pure peptide via RP-HPLC and MALDI-TOF mass spectrometry is presented in the Appendix A. For this study, we used fluconazole (FLC) as a conventional antifungal.

### 4.3. Minimum Inhibitory and Fungicidal Concentration Assays

The in vitro susceptibility of chimeric, precursor peptides and FLC was evaluated by determining the minimal inhibitory concentration (MIC) on the basis of the broth microdilution method, according to document M27-A3 of the Clinical Laboratory Standards Institute (CLSI) [39]. It was diluted in RPMI-1640 medium (Sigma-Aldrich) at concentrations ranging from 1.6 to 100 μg/mL. The initial suspension (0.5 McFarland) was mixed into saline solution. Subsequently, a 1:50 dilution in saline solution and a 1:20 dilution in RPMI-1640 were tested (0.5–2.5 × 10^3^ cells/mL). The negative control was the medium only without inoculum, and the positive control was the medium plus inoculum. The incubation time was 72 h at 30 °C at 110 rpm. The MIC values were determined by measuring the absorbance at 595 nm in a microplate reader and were defined as the lowest concentration of peptides capable of inhibiting growth equal to or higher than 50% in relation to the positive control. The MIC for fluconazole (FLC; 0.125–64 μg/mL; Pfizer, Brazil) was also determined according to M27-A3. The lowest concentration of the antifungal agent that was able to inhibit growth by 50% with respect to the positive control was considered the MIC. 

The minimal fungicidal concentration (MFC) was evaluated after the yeast’s exposure to the chimeras (1.6 to 100 μg/mL), as described above. Aliquots (3 μL) from each well of the MIC microplates were transferred to Sabouraud dextrose agar (SDA) plates and incubated at 30 °C for 74 h. The MFC was defined as the lowest peptide concentration at which ≤1 colony was visible on the agar plate.

### 4.4. Growth Inhibition and Killing Kinetics

This was evaluated for the reference strain and the clinical isolates. The inoculum was adjusted to 0.5–2.5 × 10^3^ cells/mL in RPMI-1640 medium and treated with three concentrations of each chimera (0.5 MIC, MIC, and 2 MIC). Untreated yeast cells were used as the drug-free control method. FLC was used as a conventional drug control. The suspensions were incubated in 100-well plates at 30 °C for 72 h in Bioscreen C MBR equipment, with absorbance (600 nm) automated readings taken every hour [40]. 

### 4.5. Fluorescent Staining for Yeast Viability

A commercial LIVE/DEAD™ yeast viability kit (Invitrogen, US) was used to analyze yeast viability after treatment with one of the chimeric peptides and FLC for 72 h at concentrations of 3.125, 6.25, 12.5, and 25 μg/mL and 4 and 8 μg/mL, respectively. In the dead control, the yeasts were treated with 5% hypochlorite for 10 min. Untreated yeast cells were used as a live control. Yeast cells were suspended in phosphate buffered saline (PBS). FUN-1 cell stains (10 μM) and calcofluor white (25 μM) were added to the yeast cell suspensions. After incubation in the dark at 30 °C for 30 min, the stained yeast was analyzed under a fluorescence microscope (Olympus FV1000) using filters set to excitation at approximately 488, 532, and 405 nm at 60× magnification [41]. Fungal cell viability was determined by means of fluorescence analysis in at least 10 fields. Staining and interpretation of fluorescence were performed according to the manufacturer’s instructions.

### 4.6. Checkerboard Assay

The combined effect of FLC with the chimeric peptides was evaluated for the reference strain and one clinical isolate. The amounts of 50 µL of the chimeric peptide and 50 µL of antifungal (FLC) were added to the plates at final concentrations of: 2xMIC, MIC, 1/2xMIC, 1/4xMIC, 1/6xMIC, and 1/8xMIC. A yeast cell suspension was added at 0.5–2.5 × 10^3^ cells/mL in 96-well plates. It was incubated for 72 h with shaking at 110 rpm and at 30 °C.

The effect of the combination of the two drugs was determined by calculating the fractional inhibitory concentration index (∑FIC), which was calculated as follows: ∑FIC = (MIC_combined_/MIC_alone_) FLC + (MIC_combined_/MIC_alone_) peptide. 

FIC values ≤0.5 was considered as synergistic effect, 0.5 < FIC < 1 an additive effect, 1 < FIC < 4 indifference, and FIC ≥ 4 an antagonistic effect [42]. 

### 4.7. Cytotoxicity: MTT Assay

The cell line used for this approach was the mouse fibroblast L929. The cells were washed with saline solution, trypsinized, and incubated for 5 min at 37 °C with a 5% CO_2_ atmosphere. The fibroblast cell suspension was prepared at a concentration of 1 × 10^6^ cells in 7 mL of RPMI medium supplemented with fetal bovine serum (the number of cells needed for one plate) and dispensed into 96-well plates. Then, 70 µL of cells was added per well and incubated overnight at 37 °C and 5% CO_2_. Dilutions of each peptide chimera (3.125–200 µg/mL) were formulated, and 50 µL of each were added to the plate in quadruplicate and incubated for 2 h at 37 °C and 5% CO_2_. Control wells consisted of untreated cell cultures. In addition, 10 µL of MTT was added to each of the wells, and the plate was incubated for 4 h at 37 °C with a 5% CO_2_ atmosphere. Following that, the total volume of each well was withdrawn, 100 L of DMSO was added, and it was incubated for 40 min at 37 °C with 5% CO_2_. 

Plates were read with an iMark™ microplate reader (Bio-rad, Hercules, CA, USA) measuring the absorbance at 595 nm. Wells of untreated fibroblast cells were considered to be the negative control.

### 4.8. Statistical Analysis

Curves were constructed using absorbance values and analyzed using GraphPad Prism 8.0.1 (GraphPad Software Inc., San Diego, CA, USA). Normalization of data was carried out (Shapiro–Wilk). Statistical differences were determined using analysis of variance (two-way ANOVA) followed by a Tukey–Kramer *post hoc* test. *p*-values ≤ 0.05 were considered statistically significant. 

## 5. Conclusions

The results of the present study indicate that peptide chimeras derived from LfcinB and BFII exhibit significant antifungal activity against *C. neoformans* var. *grubii*. It was established that the chemical bonding of two short sequences with low activity allows obtaining chimera with a greater antifungal effect against this yeast. The two chimeras exhibit similar antifungal activity against the tested strains, although each chimera contains a different precursor sequence with Arg substituted with Lys. No high MIC values or differences in antifungal activity of the chimeras in the mutants were found. Therefore, more studies are required to detect these chimeric peptides’ possible mechanisms of action. Furthermore, both a concentration-dependent fungistatic and fungicidal activity and a synergistic and additive effect with fluconazole were observed. Chimera cytotoxicity on murine fibroblasts showed that more than seven times the MIC is needed to kill 50% of the fibroblast cells. Therefore, these chimeras present a relevant and promising antifungal effect and can be considered to be an alternative, novel prototype with potential for future research.

## Figures and Tables

**Figure 1 antibiotics-11-01819-f001:**
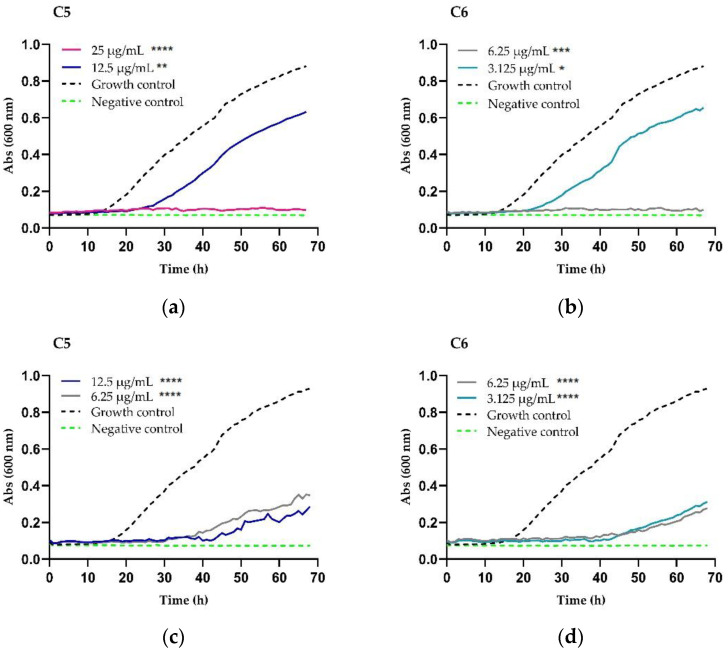
Chimeric peptides time–kill curve against *C. neoformans* var. *grubii*. (**a**,**b**) H99 strain; (**c**,**d**) clinical isolate 2807; (**e**,**f**) clinical isolate 3279; and (**g**,**h**) clinical isolate 2643. The asterisks represent the significance (* *p* < 0.05, ** *p* < 0.01, *** *p* < 0.001, **** *p* <0.0001) among the strains exposed to the different concentrations and the strains without treatment (growth control or positive control); not significant (ns). RPMI medium without yeast and peptide (sterility control or negative).

**Figure 2 antibiotics-11-01819-f002:**
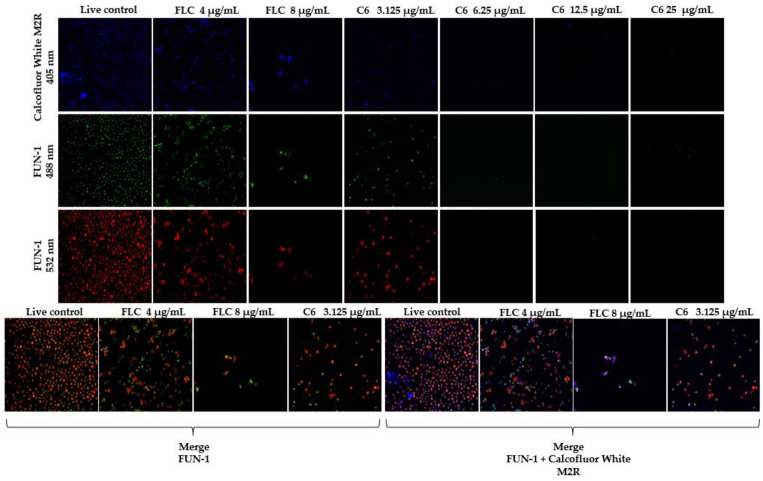
Cell viability assay of *C. neoformans* var. *grubii* after treatment (72 h) with C6 at 3.125 μg/mL, 6.25 μg/mL 12.5 μg/mL, and 25 μg/mL.

**Figure 3 antibiotics-11-01819-f003:**
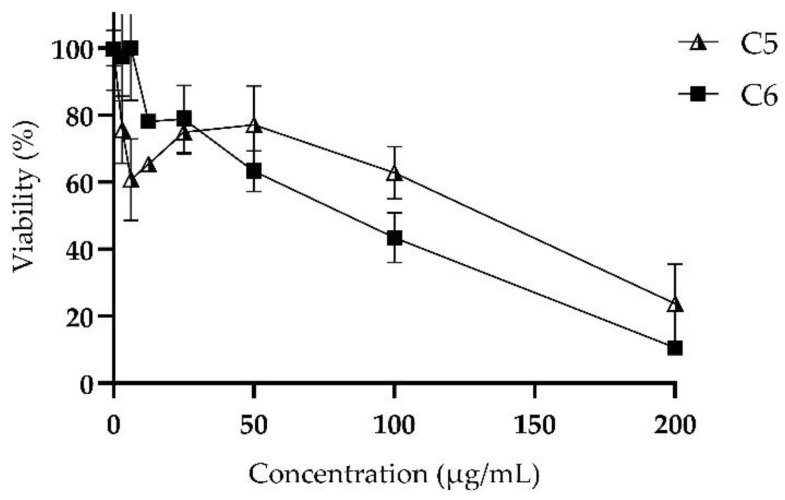
Cytotoxic effect of peptides on the mouse fibroblast cell line L929. Chimera C5: RRWQWR-Ahx-KLLKKLLK (black and white triangle shape) and chimera C6: KKWQWK-Ahx-RLLRRLLR (black box shape).

**Table 1 antibiotics-11-01819-t001:** Antifungal activity of fluconazole and peptides against *C. neoformans* var. *grubii*.

Code	Sequence	MIC/MFC µg/mL (µM)							
		H99		2807		3279		2643	
		MIC	MFC	MIC	MFC	MIC	MFC	MIC	MFC
C5	RRWQWR-Ahx-KLLKKLLK	12.5 (6)	12.5 (6)	12.5 (6)	12.5 (6)	12.5 (6)	12.5 (6)	12.5 (6)	12.5 (6)
C6	KKWQWK-Ahx-RLLRRLLR	6.25 (3)	6.25 (3)	6.25 (3)	6.25 (3)	6.25 (3)	12.5 (6)	6.25 (3)	6.25 (3)
LfcinB (20–25)	RRWQWR	>25 (>25)	>25 (>25)	>25 (>25)	>25 (>25)	>25 (>25)	>25 (>25)	>25 (>25)	>25 (>25)
BFII (32–35)_Pal_	RLLRRLLR	>50 (>46)	>50 (>46)	>50 (>46)	>50 (>46)	>50 (>46)	>50 (>46)	>50 (>46)	>50 (>46)
[K]-LfcinB (20–25)	KKWQWK	>100 (111)	>100 (111)	>100 (111)	>100 (111)	ND	ND	ND	ND
[K]-BFII (32–35)_Pal_	KLLKKLLK	>100 (114)	>100 (114)	>100 (114)	>100 (114)	ND	ND	ND	ND
FLC	-	4	-	16	-	16	-	8	-

ND: not determined. FLC: fluconazole.

**Table 2 antibiotics-11-01819-t002:** Antifungal activity of fluconazole and peptides against mutants of *C. neoformans* var. *grubii*.

Code	Sequence	MIC/MFC µg/mL (µM)																					
		CNAG_04693		CNAG_01580		CNAG_04804		CNAG_06925		CNAG_02050		CNAG_02430		CNAG_06348		CNAG_00730		CNAG_05150		CNAG_02341		CNAG_02915	
		MIC	MFC	MIC	MFC	MIC	MFC	MIC	MFC	MIC	MFC	MIC	MFC	MIC	MFC	MIC	MFC	MIC	MFC	MIC	MFC	MIC	MFC
C5	RRWQWR-Ahx-KLLKKLLK	6.25 (3)	25 (12)	≤3.125 (≤2)	3.125 (2)	3.125 (2)	3.125 (2)	6.25 (3)	6.25 (3)	6.25 (3)	6.25 (3)	6.25 (3)	6.25 (3)	6.25 (3)	6.25 (3)	6.25 (3)	6.25 (3)	12.5 (6)	12.5 (6)	12.5 (6)	12.5 (6)	≤3.125 (≤2)	3.125 (2)
C6	KKWQWK-Ahx-RLLRRLLR	6.25 (3)	12.5 (6)	6.25 (3)	6.25 (3)	6.25 (3)	6.25 (3)	6.25 (3)	6.25 (3)	6.25 (3)	6.25 (3)	6.25 (3)	6.25 (3)	6.25 (3)	12.5 (6)	3.125 (1.5)	6.25 (3)	6.25 (3)	12.5 (6)	6.25 (3)	6.25 (3)	3.125 (2)	6.25 (3)
LfcinB (20–25)	RRWQWR	>25 (>25)	>25 (>25)	>25 (>25)	>25 (>25)	>25 (>25)	>25 (>25)	>50 (>51)	>50 (>51)	>50 (>51)	>50 (>51)	>50 (>51)	>50 (>51)	>50 (>51)	>50 (>51)	>50 (>51)	>50 (>51)	>25 (>25)	>25 (>25)	>25 (>25)	>25 (>25)	>25 (>25)	>25 (>25)
BFII (32–35)_Pal_	RLLRRLLR	>50 (>46)	>50 (>46)	>50 (>46)	>50 (>46)	>50 (>46)	>50 (>46)	>50 (>46)	>50 (>46)	>50 (>46)	>50 (>46)	>50 (>46)	>50 (>46)	>50 (>46)	>50 (>46)	>50 (>46)	>50 (>46)	>50 (>46)	>50 (>46)	>50 (>46)	>50 (>46)	>50 (>46)	>50 (>46)
FLC	-	1		0.25		1		1		1		4		4		0.75		2		8		1.5	

FLC: fluconazole.

**Table 3 antibiotics-11-01819-t003:** Growth inhibition test. Antifungal activity of chimeras against *C. neoformans* var. *grubii*.

Chimera	Strain	MIC µg/mL (µM)	Effect µg/mL (µM)	
			Fungistatic	Fungicide
C5 RRWQWR-Ahx-KLLKKLLK	H99	12.5 (6)	12.5 (6)	25 (12)
	2807	12.5 (6)	6.25 (3)	>12.5 (>6)
	3279	12.5 (6)	12.5 (6)	25 (12)
	2643	12.5 (6)	6.25 (3)	>12.5 (>6)
C6 KKWQWK-Ahx-RLLRRLLR	H99	6.25 (3)	3.125 (1.5)	6.25 (3)
	2807	6.25 (3)	3.125 (1.5)	>6.25 (>3)
	3279	6.25 (3)	6.25 (3)	>12.5 (>6)
	2643	6.25 (3)	6.25 (3)	>12.5 (>6)

**Table 4 antibiotics-11-01819-t004:** Effect of combining the chimeric peptides and FLC against *C. neoformans* H99 and 2807 clinical isolate.

Synergistic Effect of Chimera/FLC Combination against *C. neoformans* var. *grubii*								
Strain	Mixture	MIC_a_	MIC_b_	A	B	FICI	MIC_a_/A	MIC_b_/B
H99	C5 + FLC	12.5	4	6.25	2	1	2	2
2807		12.5	16	0.8	4	0.3	16	4
H99	C6 + FLC	6.25	4	1.56	2	0.8	4	2
2807		6.25	16	0.4	4	0.3	16	4

FLC: Fluconazole. MICa and MICb correspond to the MIC (µg/mL) of the chimeric peptide and FLC, respectively; A and B are the MIC values of peptide and fluconazole mixture, respectively. Minimum fractional concentration index (FICI); MICa/A and MICb/B indicate how peptide or FLC are potentiated after being evaluated in combination.

## Data Availability

Not applicable.

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
