# Peer review of "In Vitro Antifungal Activity of Chimeric Peptides Derived from Bovine Lactoferricin and Buforin II against *Cryptococcus neoformans* var. *grubii"

_antibiotics, 2022, doi:10.3390/antibiotics11121819_

Round 1

Reviewer 1 Report

This study design the synthetic chimeric peptides derived from bovine lactoferricin (LfcinB) and buforin II (BFII) and evaluate their in vitro antifungal activity.  Specifically, the aim to effect the chimeras on C. neoformans var. grubii strains. All conclusions are made with reasonable experiment design but this paper has a few limitations, which should be carefully addressed.

1.  The direct correlation between chimeras and ergosterol synthesis was proved only by antifungal activity experiment, which is not convincing.

2.  Why only choose fluconazole for combination therapy effect? When combined with other antifungal agents, do chimeras have broad-spectrum combined effects?

3. Why were RRWQWR fragments of LFcinB and RLLR fragments of BFII selected instead of other fragments of these two peptides?

4. Why were these chimeric peptides not evaluated for stability? Protease stability, metal salt ion stability test? These properties are also important for peptide drug development.

5. Why didn’t the authors perform animal tests? The available data can not prove the effect of these chimeric peptides in vivo.

6. In Figure 1, it should be indicated what is the negative control. And in fact, it should be positive control.

7. In Figure 3, control groups including fluconazole and the parent peptides should be added.

8. Line 111, the sentence should be “peptide C5 was 12.5 ug/mL,a~30% decrease in growth was observed at 68-72 hours Figure 1a”.

9. Line 121-122, the sentence should be revised to “and when the peptide concentration was 3.125 to 6.25 ug/mL".

10. Line 125, the sentence should be revised to “The 3279 strain incubated with C5 at concentrations of 12.5 ug/mL exhibited a growth reduction of almost ~18%”.

Author Response

Reviewer 1:

This study design the synthetic chimeric peptides derived from bovine lactoferricin (LfcinB) and buforin II (BFII) and evaluate their in vitro antifungal activity. Specifically, the aim to effect the chimeras on C. neoformans var. grubii strains. All conclusions are made with reasonable experiment design but this paper has a few limitations, which should be carefully addressed.

Reviewer: The direct correlation between chimeras and ergosterol synthesis was proved only by antifungal activity experiment, which is not convincing.

Answer: Thanks for the comment. We would like to clarify that we did not directly assess the correlation between chimeras and ergosterol synthesis. For now, only some strains with mutations in sterol regulators were used. According with reviewer’s comment, the manuscript was modified as follows, lanes 334-335

The correlation between chimera antifungal activity and ergosterol synthesis should be evaluated, possibly using mutants directly related to ergosterol synthesis, such as ERG11 (the target of azole antifungal agents mutants) [33].

Reviewer: Why only choose fluconazole for combination therapy effect? When combined with other antifungal agents, do chimeras have broad-spectrum combined effects?

Answer: The reason for the combination of fluconazole with the chimeras is summarized in the use of this antifungal, in addition to its affordability, that it has better safety profile compared to other antifungals, and its fungistatic effect. Treatment of cryptococcosis is limited. According to the guidelines, treatment of cryptococcal meningitis is based on an induction regimen with amphotericin B and flucytosine for 1 to 2 weeks, followed by 1 week of fluconazole at high dose and 8 weeks consolidation phase of intermediate dose fluconazole and subsequently a maintenance phase with low dose fluconazole until immune reconstitution.

Reviewer: Why were RRWQWR fragments of LFcinB and RLLR fragments of BFII selected instead of other fragments of these two peptides?

Answer: According with reviewer’s comment, the follows paragraph was included, lanes 285-291

For the design and synthesis of the peptide chimeras, the minimal antimicrobial motifs reported for each Lfcin B (RRWQWR) or BFII (RLLR) were used. These motifs are the shortest sequence of the original PAM that presents activity in various microorganisms. Park, C.B. et al. synthesized a series of buforin II analogues and evaluated their antifungal activity, showing that the RLLR motif had potent antimicrobial activity [17]. Furthermore, studies have shown that the lactoferricin B active site and minimal motif includes the RRWQWR sequence [22,23].

Reviewer: Why were these chimeric peptides not evaluated for stability? Protease stability, metal salt ion stability test? These properties are also important for peptide drug development.

Answer: According with reviewer’s comment, stability tests (enzyme etc.), synthetic modifications and evaluation of their activity in new strains will be contemplated in future studies.

Reviewer: Why didn’t the authors perform animal tests? The available data cannot prove the effect of these chimeric peptides in vivo.

Answer: It's preliminary work, which includes a first in vitro screening, based on the results obtained and as mentioned above, we will move on to a new stage of study where we could consider a preliminary in vivo evaluation.

Reviewer: In Figure 1, it should be indicated what is the negative control. And in fact, it should be positive control.

Answer: According with reviewer’s comment, legend of figure 1 was modified as follows:

…without treatment (growth control or positive control); not significant (ns). RPMI medium without yeast and peptide (sterility control or negative).

Reviewer: In Figure 3, control groups including fluconazole and the parent peptides should be added.

Answer: We did not perform experiments with fluconazole because according to the reports, concentrations of 1000 µg/mL show a viability of >80% And it is not comparable with our peptides. On the other hand, for the minimal motifs of RRWQWR and RLLR, peptides were not hemolytic at concentrations below 100 µg/mL. These two peptides have not shown cytotoxic effect against fibroblasts and other non-cancerous cell lines at 200 µg/mL. For this case we only wanted to highlight the activity of the chimeras without mentioning their motives independently.

Reviewer: Line 111, the sentence should be “peptide C5 was 12.5 ug/mL, a~30% decrease in growth was observed at 68-72 hours Figure 1a”.

Answer: According with reviewer’s manuscript was modified as follows. Lane 111

“peptide C5 was 12.5 µg/mL, a ∼30% decrease in growth was observed at 68-72 hours”.

Reviewer: Line 121-122, the sentence should be revised to “and when the peptide concentration was 3.125 to 6.25 ug/mL".

Answer: Accordimg wit reviewer’s comment the manuscript was modified as follows, lane 122: when the peptide concentration was 3.125 to 6.25 µg/mL”

Reviewer: Line 125, the sentence should be revised to “The 3279 strain incubated with C5 at concentrations of 12.5 ug/mL exhibited a growth reduction of almost ~18%”.

Answer: According with reviewer’s comment, the manuscript was modified as follows, lane 125: The 3279 strain incubated with C5 at concentrations of 12.5 µg/mL exhibited a growth reduction of almost ∼18%”.

Reviewer 2 Report

This study describes the antifungal activity of antimicrobial peptides designed by merging smaller fragments with known activity. These modifications may improve the activities and physicochemical properties of AMPs.

The paper is well-written and the quality of the presentation is good. This paper is suitable for publishing in Antibiotics after addressing the following points:

·         The physicochemical properties of reported peptides should be determined and discussed to confirm that the merging really improves their drug-likeness. Since this was not the main goal of the manuscript, I suggest that authors apply some in silico methods (SwissADME and pkCSM) to predict these properties.

·         Since the antifungal activity of chimeras and their precursor sequences was determined for mutants involving TOR signaling pathway, ergosterol biogenesis, efflux pump mechanisms, authors are advised to provide initial structure-based data that may support their findings about the mechanism of action. Table S1 provides information about proteins related to target mutations. Authors may perform molecular docking of all peptides into 3D structures of these proteins that are available in PDB databank.

Author Response

Reviewer 2:

This study describes the antifungal activity of antimicrobial peptides designed by merging smaller fragments with known activity. These modifications may improve the activities and physicochemical properties of AMPs.

The paper is well-written and the quality of the presentation is good. This paper is suitable for publishing in Antibiotics after addressing the following points:

Reviewer: The physicochemical properties of reported peptides should be determined and discussed to confirm that the merging really improves their drug-likeness. Since this was not the main goal of the manuscript, I suggest that authors apply some in silico methods (SwissADME and pkCSM) to predict these properties.

Answer: As a first approach, a screening of the antifungal activity of new peptide chimeras is carried out. These results will allow us to select promising sequences that will go on to a new stage of trials where new parameters such as those mentioned by the reviewer can be determined. In silico tests such as those mentioned are proposed, as well as new stability tests (enzymatic, temperature, etc.) as well as tests in in vivo models.

Reviewer: Since the antifungal activity of chimeras and their precursor sequences was determined for mutants involving TOR signaling pathway, ergosterol biogenesis, efflux pump mechanisms, authors are advised to provide initial structure-based data that may support their findings about the mechanism of action. Table S1 provides information about proteins related to target mutations. Authors may perform molecular docking of all peptides into 3D structures of these proteins that are available in PDB databank.

Answer: Since we did not find high MIC values or differences in the antifungal activity of the chimeras in the mutants, in comparison with the reference strain and the clinical isolates, we did not carry out a structural design that provides information on possible mechanisms of action, for this, in the future we will carry out other experiments that give us more information and indications of what would be the mechanism by which the chimeras are killing the cells of C. neorformans. However, we will keep that in mind for the next study.

According with reviewer’s comment, the follows paragraph was included in the conclusions, lanes 485-491: No high MIC values or differences in antifungal activity of the chimeras in the mutants were found. Therefore, more studies are required to detect these chimeric peptides' possible mechanisms of action.

Reviewer 3 Report

The authors decided to study peptidic chimeras of the fragments of known AMPs (bovine lactoferricin and buforin II) in the context of their antifungal potency. Synthetic modification of antimicrobial peptides is very promising and widely applied strategy in the design of novel compounds with enhanced activity against pathogens.

The manuscript is thoroughly prepared and well written. It should be highlighted, that results are clearly summarized and presented in tables, which help the readers understand step by step the differences in activities of chimeras and fragments of native peptides.

In my opinion, each part of presented research was planned properly. However, I would like to ask the authors, why did they choose this kind of connection of LfcinB and BFII fragments? Do they plan to expand their studies with chimeras like KLLKKLLK-Ahx-RRWQWR, and RLLRRLLR-Ahx-KKWQWK? 6-aminohexanoic acid is a compound widely used as a spacer, e.g. in coupling biotin or fluorophores to the peptides. The lenght of the hydrocarbon chain might have the influence on the antimicrobial activity of studied compounds. Did the authors apply any other spacers in their chimeras?

Considering any peptides as a potential treatment agents, stability studies of those compounds should be included in the research plan. Therefore, I highly recommend the authors to perform this kind of experiments for newly designed peptide chimeras described in the manuscript.

Author Response

Reviewer 3:

The authors decided to study peptidic chimeras of the fragments of known AMPs (bovine lactoferricin and buforin II) in the context of their antifungal potency. Synthetic modification of antimicrobial peptides is very promising and widely applied strategy in the design of novel compounds with enhanced activity against pathogens.

The manuscript is thoroughly prepared and well written. It should be highlighted, that results are clearly summarized and presented in tables, which help the readers understand step by step the differences in activities of chimeras and fragments of native peptides.

Reviewer: In my opinion, each part of presented research was planned properly. However, I would like to ask the authors, why did they choose this kind of connection of LfcinB and BFII fragments? Do they plan to expand their studies with chimeras like KLLKKLLK-Ahx-RRWQWR, and RLLRRLLR-Ahx-KKWQWK? 6-aminohexanoic acid is a compound widely used as a spacer, e.g. in coupling biotin or fluorophores to the peptides. The lenght of the hydrocarbon chain might have the influence on the antimicrobial activity of studied compounds. Did the authors apply any other spacers in their chimeras?

Answer: According with reviewer’s comment, the follows paragraph was included, lanes 285-291

For the design and synthesis of the peptide chimeras, the minimal antimicrobial motifs reported for each Lfcin B (RRWQWR) or BFII (RLLR) were used. These motifs are the shortest sequence of the original PAM that presents activity in various microorganisms. Park, C.B. et al. synthesized a series of buforin II analogues and evaluated their antifungal activity, showing that the RLLR motif had potent antimicrobial activity [17]. Furthermore, studies have shown that the lactoferricin B active site and minimal motif includes the RRWQWR sequence [22,23].

Reviewer: Why did they choose this kind of connection of LfcinB and BFII fragments?: Do they plan to expand their studies with chimeras like KLLKKLLK-Ahx-RRWQWR, and RLLRRLLR-Ahx-KKWQWK?  Did the authors apply any other spacers in their chimeras?

Answer: Previous studies showed that chimeras containing these two motifs had greater antibacterial activity, as well as the inclusion of the Ahx spacer facilitated synthesis and also increased antibacterial activity.

Reviewer: Considering any peptides as a potential treatment agents, stability studies of those compounds should be included in the research plan. Therefore, I highly recommend the authors to perform this kind of experiments for newly designed peptide chimeras described in the manuscript.

Answer: We agree with reviewer’s recommendation, For the next step of this study, we will first consider evaluating the stability of the peptides.

Round 2

Reviewer 1 Report

The authors have carefully revised the manuscript according to the suggestion. I have no more comments.